# No pre-zygotic isolation mechanisms between *Schistosoma haematobium* and *Schistosoma bovis* parasites: From mating interactions to differential gene expression

Julien Kincaid-Smith[1,2‡], Eglantine Mathieu-Bégné[1‡]*, Cristian Chaparro[1], Marta Reguera-Gomez[3], Stephen Mulero[1], Jean-Francois Allienne[1], Eve Toulza[1], Jérôme Boissier[1]*

1 Univ. Montpellier, CNRS, IFREMER, UPVD, IHPE, Perpignan, France, 2 Centre for Emerging, Endemic and Exotic Diseases (CEEED), Department of Pathobiology and Population Sciences (PPS), Royal Veterinary College, University of London, Hawkshead Campus, Herts, United Kingdom, 3 Departamento de Parasitología, Facultad de Farmacia, Universidad de Valencia, Burjassot, Valencia, Spain

‡ These authors share first authorship on this work.
* eglantine.mb@gmail.com (EM-B); Boissier@univ-perp.fr (JB)

## Abstract

Species usually develop reproductive isolation mechanisms allowing them to avoid inter-breeding. These preventive barriers can act before reproduction, "pre-zygotic barriers", or after reproduction, "post-zygotic barriers". Pre-zygotic barriers prevent unfavourable mating, while post-zygotic barriers determine the viability and selective success of the hybrid off-spring. Hybridization in parasites and the underlying reproductive isolation mechanisms maintaining their genetic integrity have been overlooked. Using an integrated approach this work aims to quantify the relative importance of pre-zygotic barriers in *Schistosoma haematobium x S. bovis* crosses. These two co-endemic species cause schistosomiasis, one of the major debilitating parasitic diseases worldwide, and can hybridize naturally. Using mate choice experiments we first tested if a specific mate recognition system exists between both species. Second, using RNA-sequencing we analysed differential gene expression between homo- and hetero-specific pairing in male and female adult parasites. We show that homo- and hetero-specific pairing occurs randomly between these two species, and few genes in both sexes are affected by hetero-specific pairing. This suggests that i) mate choice is not a reproductive isolating factor, and that ii) no pre-zygotic barrier except spatial isolation "by the final vertebrate host" seems to limit interbreeding between these two species. Interestingly, among the few genes affected by the pairing status of the worms, some can be related to pathways affected during male and female interactions and may also present interesting candidates for species isolation mechanisms and hybridization in schistosome parasites.

**Data Availability Statement:** The authors confirm that all data underlying the findings are fully available without restriction. Raw sequences used in this study have been submitted to the Sequence Read Archive under the BioProject PRJNA491632, the transcriptome assembly have been submitted on the Figshare repository (https://doi.org/10.6084/m9.figshare.12581156).

**Funding:** This work has been funded by the French Research National Agency (project HySWARM, grant no. ANR-18-CE35-0001). SM was supported by the Occitania region (project MOLRISK, award no. NREST 2019/1/059), the European "Fonds Européen de Développement Régional" (FEDER) and MRG by the Fellowship of "Estancias breves" (linked to the Programa de Ayudas de Formacion de Profesorado Universitario 2015, Ministerio de Ciencia, Innovación y Universidades, Spain, https://www.educacionyfp.gob.es/servicios-al-ciudadano/catalogo/general/20/200487/ficha/200487-2017.html#dc1). The funders had no role in study design, data collection and analysis, decision to publish, or preparation of the manuscript.

**Competing interests:** The authors have declared that no competing interests exist.

## Author summary

Understanding how species maintain their genetic integrity is a central question in evolutionary biology. While isolation mechanisms are well documented in free-living organisms, it is currently not the case for parasite species. Yet, occurrence of parasite hybrids is a critical global health concern since these hybrids are expected to be more harmful than parental species. We addressed the question of reproductive isolation mechanisms in parasitic species by conducting an integrative experimental study (from mate choice to gene expression) on two schistosome species (*Schistosoma haematobium* and *S. bovis*) that parasitize human and cattle, respectively. Importantly, their hybrid progeny has been involved in recent outbreaks, including outbreaks outside of endemic areas. We showed that rather than having a homo-specific mate choice, *S. haematobium* and *S. bovis* mate randomly. Also, male and female worms only express a few genes differentially when involved in a hetero-specific pair compared to a homo-specific pair. We consequently suggest that these two schistosome species lack strong reproduction isolation mechanisms, except those imposed by specificity to the final host species. Our results raise the concern that in the absence of post-zygotic barriers in sympatric zones hybridization might be more common than previously thought if these two species are able to encounter each other.

## Introduction

A subset of obstacles evolved in the course of speciation in order to limit gene flow via hybridization and maintain species boundaries. These obstacles are traditionally classified as pre- and post-zygotic barriers (also known as pre- or post-mating barriers) and can be defined as any mechanism preventing or reducing gene flow between groups of potentially interbreeding individuals [1]. Pre-zygotic barriers include spatial isolation (*e.g.*, two species live in different habitats), behavioural isolation (*e.g.*, individuals can choose to mate with individuals of their own species), temporal isolation (reproduction does not occur at the same time *e.g.*, different seasons), mechanical isolation (sex organs are not compatible) and gametic isolation (sperm and eggs mix but fertilization does not occur). When the first barrier is crossed, post-zygotic isolation mechanisms can arise to prevent gene flow. Post-zygotic barriers include hybrid unviability (hybrids die prematurely), reduced fitness with low fertility (hybrids are less fertile, infertile or non-viable) or hybrid breakdown (a longer process where the hybrid lines are counter-selected compared to their parental forms). The strength and/or the order of each reproductive barrier vary among species. This makes difficult to predict the outcome of inter-species mating, and the evolution of reproductive isolation mechanisms [2]. Moreover, reproductive isolation is often the result of an accumulation and interaction of multiple pre- and post-zygotic mechanisms restricting most gene flow [3]. However, it is generally recognized that pre-zygotic isolation barriers are enhanced in sympatric species [4], and are the most effective because they act early to prevent the production of hybrid progeny.

Despite their importance in terms of biodiversity [5], but also animal and human health, parasite species have received less attention than other free-living organisms regarding both hybridization and the role of reproductive isolation mechanisms [6]. Pre-zygotic barriers in parasites usually include additional and stronger obstacles to overcome compared to those of free-living organisms. For instance, the "habitat barrier" includes the geographic area, the host species and the tropism within the host. For parasites, hosts are dynamic habitats imposing strong selective pressures (co-evolutionary arms race) requiring constant adaptation of

parasites for the completion of their life cycle. The specialisation of parasite species to a particular host is thus expected to be a strong pre-zygotic isolation mechanism preventing hybridization and favouring speciation. However, some closely related species do manage to retain their genetic identity whilst parasitizing the same host, meaning that they have acquired selective mechanisms for reproductive isolation. Hybridization and pre-zygotic reproductive barriers have been studied on very few parasite models such as plasmodium species, cestodes and schistosomes [6–8]. Partial pre-zygotic barriers have been evidenced between *Plasmodium berghei* and *P. yoeli* [7]. It was not the case between *Schistocephalus solidus and S. pungitii* [6], suggesting in the latter that post-zygotic selection against hybridization is presumably the most important driving force limiting gene flow between these two parasitic sister species [6].

Schistosomes are parasitic agents that cause schistosomiasis, a debilitating disease affecting over 240 million people worldwide, mainly in tropical and subtropical areas [9]. There are currently 23 know species in the genus *Schistosoma*, including six species that infect humans and 20 species that infect animals [8]. These parasites have a two-host life cycle, which includes a mammalian definitive host, in which sexual reproduction occurs and a mollusc intermediate host in which asexual multiplication takes place. Schistosomes have the particularity of having separate sexes, a feature not observed in other trematodes that are hermaphroditic [10,11]. Schistosomes have therefore been intensively studied for their sexual features including male-female interactions [12,13], sex-ratios [14,15], mating systems [16,17] and mating behaviour [18]. One direct consequence of dioecism in these species is the necessity of individuals of both sexes to infect the same definitive host. This constraint can lead to interactions between species infecting the same host, and in the case of porous reproductive pre-zygotic barriers this can lead to hybridization.

To conserve their genetic identity, schistosomes that inhabit the same definitive host are expected to present pre-zygotic isolation mechanisms. Among these barriers, habitat and behavioural isolation have a great influence in schistosome's sexual interactions. First, habitat isolation is a three-level constraint that initially has to be overcome (*i.e.*, same geographic area, same host individual, and same localisation in the host). Indeed, schistosomes species are distributed worldwide (the majority in Africa), the vertebrate host specificity depends on the parasite species, and while the majority of species live in the mesenteric vein system, one species (*S. haematobium*) lives in the veins surrounding the bladder of humans. Second, behavioural isolation is more complex in schistosomes than in other species because mating is followed by a pairing-dependent differentiation of the female's sexual organs [12,13]. Studies have clearly established that the presence of the male (independently of the species paired) is necessary not only for the female's sexual development, but also for the maintenance of a sexually mature and active state [19–21]. It was also demonstrated that female schistosomes stimulate males through changes in levels of glutathione and lipids, and stimulate tyrosine uptake in the male worms [12]. Hence, while males transfer glucose and lipid secretions to females, females also release factors affecting the physiology of male worms [22–25]. Thus, male and female schistosomes are strongly co-dependent, in terms of behaviour (*i.e.*, they have complementary roles in the hosts), but also physiologically [10] with an intimate and permanent association between sexes necessary for reproduction to occur.

Nevertheless, several hybrid schistosomes have been evidenced [8,22,26,27]. Similarly to other groups, isolation mechanisms increase with divergence time between taxa [4,8]. The success of inter-species interactions on the viability of hybrid offspring also depends on the direction of the cross and thus which parental species provides the maternal and paternal genome [27–29]). Studies on schistosome mate choices have revealed that depending on the parasite species interacting, some combinations may readily pair with no preference (*S. haematobium x S. intercalatum* (referred to as *S. guineensis* since 2003 based on their mitochondrial

divergence [30,31]) *S. bovis x S. curassoni* and *S. mansoni x S haematobium*), whereas when involved in other combinations, species may present a mate recognition system favoring or not interspecies pairing (*S. mansoni x S. intercalatum* (now *S. guineensis*), *S. haematobium* x *S. mattheeii*, and *S. mansoni x S. margrebowiei* crosses) [28,32–34]. However, competition between schistosome species can also explain the frequency of some interspecific crosses [28,29,32]. For instance it has been shown that *S. haematobium* males can take away females from other species when competing with male *S. intercalatum* (now *S. guineensis*) [35], S. *mattheei* [29] or *S. mansoni* [28] hence promoting or favouring hetero-specific pairing. For schistosome species that randomly pair with no mate preference and for many related parasitic species capable of hybridizing, final host specificity may be the sole barrier preventing interbreeding [35]. This isolation mechanism "by the host" may be so efficient that species may lack any post-zygotic or other pre-zygotic mechanisms ultimately allowing them to hybridize when the opportunity arises. Therefore, the lack of reproductive incompatibility (*i.e.*, isolation by behaviour and physiology) between schistosome species infecting humans and animals may facilitate gene flow if the host isolation barriers are broken down.

*Schistosoma haematobium x S. bovis* hybrids are today the most studied hybrid system of schistosomes. These hybrids were first identified in Niger by Brémont [36] and more recently in Senegal [26] but appeared widely distributed in West Africa [26,27,36–39]. Moreover, these hybrids have recently been involved in a large-scale outbreak in Europe (Corsica, France), where transmission of the disease is persistent [37,40]. *Schistosoma haematobium* and *S. bovis* are co-endemic in Africa, but their host specificity and tropism within their definitive hosts are different (urogenital and human vs. intestinal and cattle, respectively). *S. haematobium* is mainly a parasite of humans, however, sporadic studies have shown that non-human primates, *Cetartiodactyla* members or rodents could be naturally infected by this parasite species (although these accounts were based on egg morphology and could thus involve other species) [41–43]. Conversely, *S. bovis* is mainly a parasite of ruminants with sporadic cases of rodent infection [41,44]. Interestingly, although data are scarce, recent studies showed that *S. haematobium* x *S. bovis* hybrids may naturally infect rodents or cattle [39,44].

Hybridization between these two species is particularly worrying because it raises the eventuality for a human parasite to have animal reservoirs of infection and the animal parasite to be zoonotic [45]. Likewise, hybridization may lead to changes in the parasites life history traits, including host range expansion, increased virulence and host morbidity, but also response to chemotherapeutic treatment [46]. Indeed, in experimental infections, these hybrids often display heterosis, in which their fitness outperforms the fitness of parental species [8,26,27]. Importantly the existence of a mate recognition system between the two species would prevent natural occurrences of hybridization in sympatric areas. In contrary a lack of reproductive isolation could indicate that occurrences of hybridization may be more frequent.

Although experimental crosses in hamsters have demonstrated their capacity to pair and the viability of *S. haematobium* x *S. bovis* hybrids [27], their pairing frequency and underlying molecular mechanisms need to be assessed. This study hence uses an integrated approach, from mating behaviour to male and female gene expression, in order to quantify the importance of pre-zygotic barriers involved in the interactions between *S. haematobium x S. bovis*. First, using a mate choice experiment we tested whether specific mate recognition or competition exists by quantifying the frequency of hetero-specific and homo-specific pairs compared to random mating expectations. Second, given the strong co-dependence between male and female schistosomes, we also analysed the influence of pairing (homo- vs hetero-specific) on the transcriptomic profile of male and female parasites using RNA sequence analysis. We hypothesize that since these hybrids are frequently encountered in the field [37,37,39] and since parental species are able to pair in the laboratory [27] mate recognition should not

constitute a strong barrier to reproduction. However, depending on species dominance in mating, the direction of pairing could be affected. Since females undergo strong developmental changes upon pairing [13,47,48] we would expect finding strong transcriptomic changes associated with inter-species interactions for females but not for males. The molecular determinants of the very first step towards hybridization may give further insight into the permeability of the two species and reveal some important genes linked to male and female interaction, species isolation and hybridization.

## Materials and methods

### Ethics statement

Experiments were carried out according to national ethical standards established in the writ of February 1st, 2013 (NOR: AGRG1238753A), setting the conditions for approval, planning and operation of establishments, breeders and suppliers of animals used for scientific purposes and controls. The French Ministry of Agriculture and Fishery (Ministère de l'Agriculture et de la Pêche), and the French Ministry for Higher Education, Research and Technology (Ministère de l'Education Nationale de la Recherche et de la Technologie) approved the experiments carried out for this study and provided permit A66040 for animal experimentation. The investigator possesses the official certificate for animal experimentation delivered by both ministries (Décret n° 87–848 du 19 octobre 1987; number of the authorization 007083).

### Origin and maintenance of schistosome strains

*Schistosoma haematobium* and *S. bovis* were maintained in the laboratory using *Bulinus truncatus* snails as intermediate hosts and *Mesocricetus auratus* as definitive hosts. The parasite strains originated from Cameroon and Spain for *S. haematobium* and *S. bovis*, respectively [49]. The *S. haematobium* strain was initially recovered from the urine of infected patients in 2015 (Barombi Kotto lake; 4°28'04"N, 9°15'02"W). Eggs from positive samples were hatched, *miracidia* were harvested, and sympatric *B. truncatus* molluscs, bred from snails collected from the same location as the parasites, were individually exposed to five *miracidia* before being transferred to the IHPE laboratory for their maintenance. The *S. bovis*, strain isolated in the early 80's [49,50] was kindly provided by Ana Oleaga from the Spanish laboratory of parasitology of the Institute of Natural Resources and Agrobiology in Salamanca, and originates from Villar de la Yegua-Salamanca.

**Experimental infections.** Protocols of experimental infections were set for two objectives, i) quantifying the frequency of hetero-specific and homo-specific pairings and, ii) forcing hybridization and then assessing the transcriptomic changes between homo-specific and hetero-specific paired males and females. The successive steps of our experimental infection procedure are presented in Fig 1. Detailed procedures for mollusc and rodent infections have been previously described [51–53]. Step 1: 3–5 mm *B. truncatus* were placed in 24-well plates containing 1ml of spring water per well. Each mollusc was exposed overnight to a single *miracidium* (*i.e.*, a single male or female genotype) of either *S. haematobium* or *S. bovis*. The following morning molluscs were placed in breeding tanks and fed *ad libitum* for the duration of the experiment. After a minimum period of 55 days, corresponding to the development time of the parasites in their intermediate host, molluscs were stimulated under light for cercariae shedding. Step 2: cercariae from each infected mollusc were recovered for molecular sexing as previously described [49]. Step 3: molluscs were gathered into four distinct tanks according to the species and the sex of the infecting parasite. Step 4: hamsters were individually exposed to *cercariae* using the surface application method for one hour [51–53]. The sex and the species of the *cercariae* used for each experiment are presented in Table 1 and are described further below (*Mate choice analysis, Force pairing and underlying molecular analysis*).

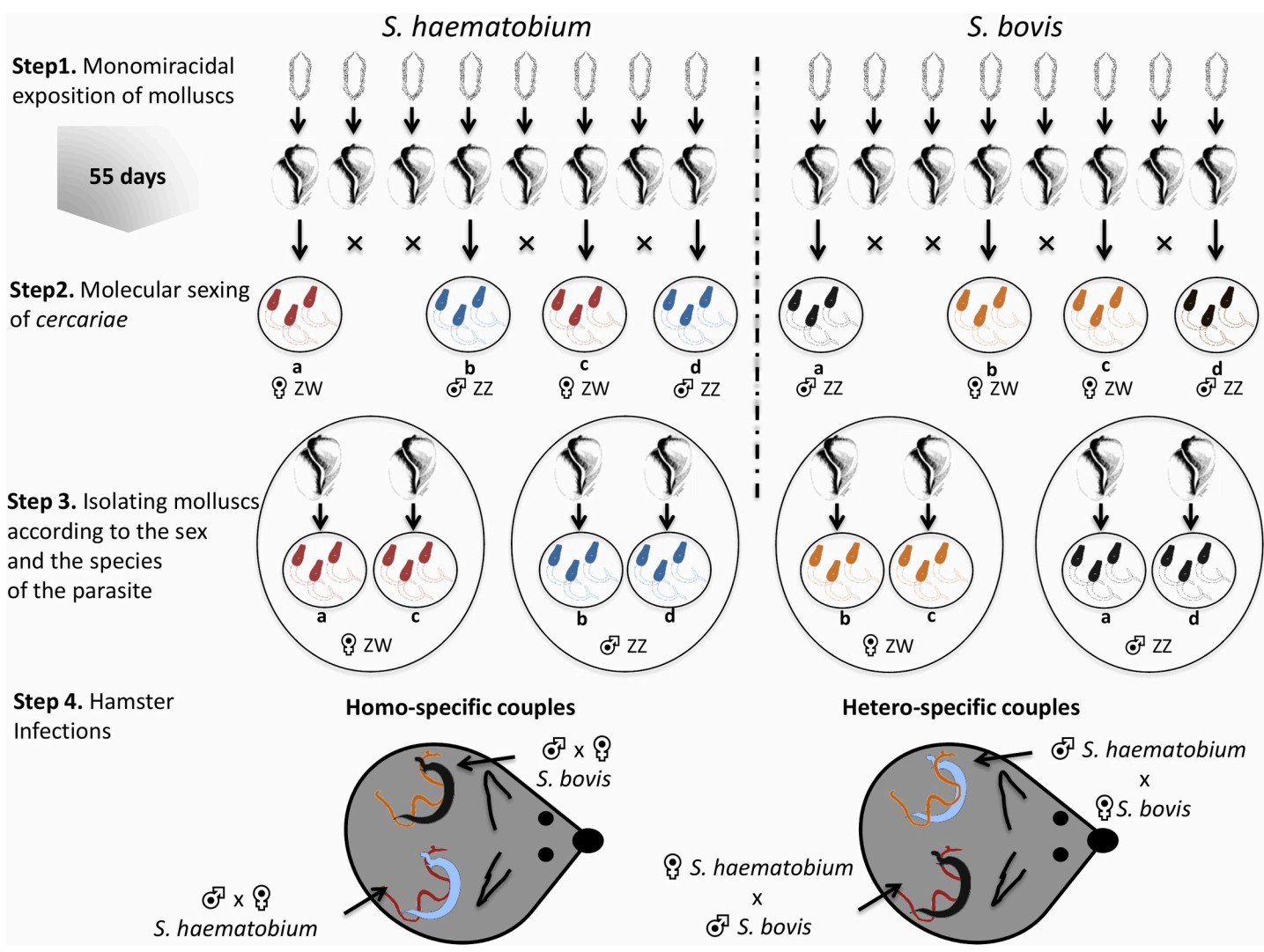

**Fig 1. Schematic representation of experimental infection procedure.**

Hamsters were euthanized at three months after cercarial exposure and adult worms were recovered by hepatic perfusion [53]. Hamsters were autopsied and specific organs such as the mesenteric and portal veins were carefully checked to identify potential remaining worms. We recorded each worm's sex inferred by their strong sexual dimorphism [11] and their paring status (paired or single). Paired worms were manually separated under a light microscope. All worms collected were individualized in 96-well plates and were subjected to DNA extraction using the method described previously in Beltran et al. (2008) [54]. The species of each worm was identified using the rapid diagnostic procedure based on multiplex PCR reaction described by Webster and colleagues [55,56].

## Mate choice analysis

**Experimental design.** The experimental procedure to quantify the frequency of homo- and hetero-specific pairs between *S. bovis* and *S. haematobium* consisted of five experiments (*i. e.*, Exp. 1 to Exp. 5, see Table 1). The first four experiments aimed to test individually the

**Table 1. Number of cercariae used for each experiment according to the species and the sex of the parasite.**

| Experiments | S. haematobium | | S. bovis | | Number of hamsters |
|---|---|---|---|---|---|
| | Males | Females | Males | Females | |
| **Objective 1:** Quantification of homo- and hetero-specific pairs frequency | | | | | |
| **Limited Choice Experiment** | | | | | |
| Exp. 1 (limiting sex: *S. haematobium* males) | 150 | 225 | - | 225 | 5 |
| Exp. 2 (limiting sex: *S. haematobium* females) | 225 | 150 | 225 | - | 5 |
| Exp. 3 (limiting sex: *S. bovis* males) | - | 225 | 150 | 225 | 5 |
| Exp. 4 (limiting sex: *S. bovis* females) | 225 | - | 225 | 150 | 5 |
| **Full Choice Experiment** | | | | | |
| Exp. 5 | 150 | 150 | 150 | 150 | 5 |
| **Objective 2:** Assess the transcriptomic profiles of homo- and hetero-specific paired worms | | | | | |
| Homo-specific forced pairing 1 | 300 | 300 | - | - | 6 |
| Homo-specific forced pairing 2 | - | - | 300 | 300 | 6 |
| Hetero-specific forced pairing 1 | 300 | - | - | 300 | 6 |
| Hetero-specific forced pairing 2 | - | 300 | 300 | - | 6 |

choice of each species and sex (Exp. 1 and Exp. 3 for male choice—Exp. 2 and Exp. 4 for female choice for *S. haematobium* and *S. bovis*, respectively). In each experiment, five hamsters (used as biological replicates) were infected with mixed combinations of cercariae (Table 1). These four experiments represented a limited choice of mate where excess of one sex (of both species competing for mating) ensuring that all individuals of the other sex (that had the choice for homo- or hetero-specific pairings) will be mated (Table 1). Finally, the last experiment (Exp. 5, Table 1) represented full choice of mate. Hamsters were infected with equal numbers of cercariae of both sexes and both species so that all combination of mating can be assessed at the same time.

**Statistical analysis.** After counting the total number of adult worms recovered for each species (*e.g.*, homo-specific pairs, hetero-specific pairs and single worms), we calculated the expected number of single and paired worms according to the null hypothesis of random pairing (*e.g.*, in the Exp. 1, the expected number of homo-specifically paired *S. haematobium* males equals the total number of *S. haematobium* males, times the total number of *S. haematobium* females over the total number of females). Expected and observed numbers of homo- and hetero-specific pairs were then compared using Chi-square tests.

### Forced pairing and underlying molecular analysis

**Experimental design.** Hamsters were infected with four combinations of parasites (Table 1). Homo-specific pairing consisted of infections with single species of cercariae, while hetero-specific pairing consisted of infections with male and female cercariae of the opposite species. Hamsters were euthanized three months after their exposition to cercariae and adult worms were recovered by hepatic perfusion. Paired worms were separated under a magnifier using a small paintbrush (to avoid causing any damage to the worms) and pooled according to their sex (male or female) and the species of their sexual partner (same or opposite species). Pools of 10–12 female or male worms were placed in 2ml microtubes and immediately frozen in liquid nitrogen and stored at -80°C. Three biological replicates were constituted for each combination representing a total of 24 samples (2 sexes x 4 combinations x 3 replicates) for subsequent RNA extraction and transcriptome sequencing (see Fig 2 for a schematic view of the procedure).

***RNA extraction and transcriptome sequencing of homo- and hetero-specific* S. haematobium *and* S. bovis *male and female pairs.*** Trizol RNA extraction and subsequent paired-end Illumina HiSeq 4000 PE100 sequencing technology was performed on the 24 samples. Briefly, pools of adult worms were ground with two steel balls using a Retsch MM400 cryo-brush (2 pulses at 300Hz for 15s). Total RNA was extracted using the Trizol Thermo Fisher Scientific protocol (ref: 15596018) slightly modified as the volume of each reagent was halved. Total RNA was eluted in 44 µl of ultrapure water before undergoing a DNase treatment using Thermofisher Scientific Turbo DNA-free kit. RNA was then purified using the Qiagen RNeasy mini kit and eluted in 42µl of ultrapure water. Quality and concentration of the RNA was assessed by spectrophotometry with the Agilent 2100 Bioanalyzer system and using the Agilent RNA 6000 nano kit. Further details are available at Environmental and Evolutionary Epigenetics Webpage (http://methdb.univ-perp.fr/epievo).

**Illumina library construction and high-throughput sequencing.** cDNA library construction and sequencing were performed at the Génome Québec platform. The TruSeq stranded mRNA library construction kit (Illumina Inc., USA) was used following the manufacturer's protocol on 300 ng of total RNA per sample. Sequencing of the 24 samples was performed in 2x100 bp paired-end on a Illumina HiSeq 4000 (S1 Table). Sequencing data are available at the NCBI-SRA under the BioProject PRJNA491632.

***Transcriptomic analysis of hetero-specific pairing* versus *homo-specific pairing.*** Raw sequencing reads were analysed on the Galaxy instance of the IHPE laboratory [57,58] First, raw reads were subjected to quality assessment and sequence adaptor trimming. We used the set of tools based on the FASTX-toolkit [59], as well as Cutadapt program (Galaxy Version 1.16.1) to remove adapter sequences from Fastq files [60]. Finally, paired end reads were joined in a single fastq file using the FASTQ interlacer/de-interlace programs (Galaxy Version 1.1). Processed reads were mapped using RNA-star Galaxy Version 2.6.0b-1 [61] to the *S. haematobium* reference genome [62] downloaded from the *Schistosoma* Genomic Resources website SchistoDB (http://schistodb.net/common/downloads/Current_Release/ShaematobiumEgypt/fasta/data/). Exon-intron structure was thereafter reconstructed for each mapping BAM file using Cufflinks transcript assembly Galaxy Version 2.2.1.2, by setting the max intron length at 50000, but without any correction parameters [63]. Finally, in order to create a reference transcriptome representative of *S. haematobium* and *S. bovis* male and female reads, we merged all cufflinks data with Cuffmerge Galaxy Version 2.2.1.2 [63] without using any guide or reference. This enabled us to create a representative reference transcriptome of both species and both sexes using the same reference genome. The Genomic DNA intervals of all newly assembled genes of this reference transcriptome were extracted from the *S. haematobium* reference genome and converted into a Fasta file.

The number of reads per transcript for each sample (*i.e.*, the read abundance representative of each gene) was quantified using HTseq-count Galaxy Version 0.9.1 on the reference transcriptome, setting the overlap resolution mode on "union" [64]. Finally, we evaluated the differential gene expression levels between homo-specifically and hetero-specifically paired worms for each species and each sex separately using DESeq2 Version 1.28.1 [65] run on R version 4.0.0 [66]. We carried out four types of comparisons which respectively focused on *S. haematobium* males, *S. haematobium* females, *S. bovis* males and *S. bovis* females and contrasted gene expression profiles between hetero-specifically paired individuals and homo-specifically paired individuals. Differential gene expression results were filtered on the adjusted *P-value* (Benjamini-Hochberg multiple testing based False Discovery Rate (FDR)) and considered significant when $\leq 5\%$.

**Functional annotation.** Using our BLAST local server, we annotated the entire *de novo* assembled transcriptome by Blastx search against the non-redundant database of the NCBI.

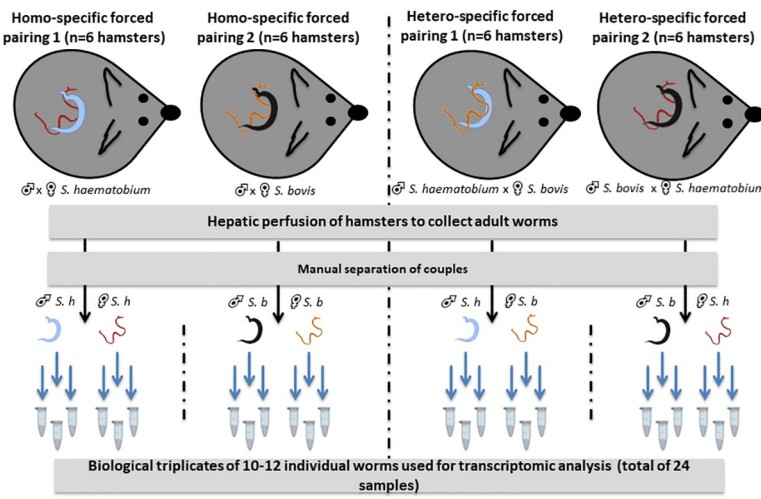

S. h = S. haematobium
S. b = S. bovis

**Fig 2. Schematic representation of the procedure used to obtain the reciprocal homo- and hetero-specific pairs of S.** *haematobium* and *S. bovis.*

We conserved only the longest unique transcript (TCONS) of each representative gene (XLOC) for Blastx search and subsequent analysis. Output XML files were used for gene ontologies (GO) mapping and annotation using Blast2Go version 4.1.9 [67]. Finally, enrichment Fisher's exact tests were performed on up and down regulated sets of genes focusing on biological process (BP) ontology terms. The P-value for significance was set to 5% False Discovery Rate (FDR).

## Results

### Mating choice experiments

**Limited choice: Experiments 1 to 4.** Details on the number of worms recovered from each hamster and whether they were paired or single are summarized in Table 2. For each mate choice experiment both homo-specific and hetero-specific pairs were observed (Table 2, Exp. 1–4). Also, in each limited choice of mate experiment (Exp. 1–4) we consistently obtained an excess of single worms of both species competing for pairing (*i.e.*, male or female depending on the experiment) whereas all worms of the limiting sex (*i.e.*, choosing partners, such as female choice or male competition) were paired (Table 2). This indicates that the choosing partners in each experiment were not limited in their choice by the number of potential homo- or hetero-specific partners. Specifically, in the experiment 1, the number of homo- and hetero-specific pairs of male *S. haematobium* was significantly different from those expected under the random mating hypothesis ($\chi2 = 11.10$; d.f. = 4; P-value = 0.049, Table 2). This was due to the deviation from the random mating hypothesis in one hamster (hamster number 3, see Table 2). Regarding *S. haematobium* females' choice (Table 2, Exp. 2) at the contrary, the numbers of homo-specific pairs and hetero-specific pairs were not significantly different from expectations under the random mating hypothesis ($\chi2 = 3.118$; d.f. = 4; P-value = 0.682, Table 2). In the experiment 3 that focused on *S. bovis* males' choice, the total number of paired worms recovered was extremely low, due to premature death of two hamsters and only two hamsters had enough worms to be analysed (Table 2, Exp. 3). Although in this case, statistics should be interpreted with caution, the numbers of homo-specific and hetero-specific pairs

**Table 2. Summarized information of experiments 1 to 4 (limited choice).** For each experiment are displayed the sex and the species of the choosing partner (such as female choice or male competition), the number of observed homo- and hetero-specific pairs and the number of worms that remained single. Sh = *S. haematobium* and Sb = *S. bovis*. Expected number of pairs under random mating hypothesis is shown in brackets (see the statistics section in Materials and Methods for details). Chi-square statistic, degree of freedom and P-value are given for each hamster, for each experiment and for all experiments combined. * indicates significant results at 5% level. In Exp. 3, worms from only two hamsters could be analysed, while others died prematurely (two hamsters) or presented too few numbers of paired worm (one hamster).

| Exp. | Host | Choosing partner | Homo-specific pairs | Hetero-specific pairs | Single worms | | $\chi^2$-statistic | d.f. | P-value |
|---|---|---|---|---|---|---|---|---|---|
| **Exp. 1** | | | ♂ Sh x♀ Sh | ♂ Sh x♀ Sb | ♀ Sh | ♀ Sb | **11.104** | **4** | **0.049*** |
| 1 | 1 | ♂ Sh | 11 (14) | 14 (11) | 20 | 9 | 1.838 | 1 | 0.175 |
| 1 | 2 | ♂ Sh | 11 (15) | 22 (18) | 15 | 11 | 1.543 | 1 | 0.214 |
| 1 | 3 | ♂ Sh | 14 (20) | 16 (10) | 25 | 4 | 5.057 | 1 | 0.025* |
| 1 | 4 | ♂ Sh | 9 (9) | 6 (6) | 36 | 26 | 0.015 | 1 | 0.903 |
| 1 | 5 | ♂ Sh | 10 (13) | 10 (7) | 41 | 15 | 2.651 | 1 | 0.103 |
| **Exp. 2** | | | ♀ Sh x ♂ Sh | ♀ Sh x ♂ Sb | ♂ Sh | ♂ Sb | **3.118** | **4** | **0.682** |
| 2 | 1 | ♀ Sh | 10 (9) | 2 (4) | 7 | 5 | 0.908 | 1 | 0.341 |
| 2 | 2 | ♀ Sh | 6 (5) | 1 (2) | 0 | 1 | 0.429 | 1 | 0.513 |
| 2 | 3 | ♀ Sh | 12 (10) | 3 (5) | 11 | 10 | 1.688 | 1 | 0.194 |
| 2 | 4 | ♀ Sh | 16 (15) | 3 (4) | 6 | 2 | 0.094 | 1 | 0.759 |
| 2 | 5 | ♀ Sh | 12 (12) | 13 (13) | 11 | 12 | 0.001 | 1 | 0.993 |
| **Exp. 3** | | | ♂ Sb x♀ Sb | ♂ Sb x♀ Sh | ♀ Sb | ♀ Sh | **4.522** | **1** | **0.104** |
| 3 | 2 | ♂ Sb | 4 (2) | 0 (2) | 46 | 55 | 4.400 | 1 | 0.036 |
| 3 | 3 | ♂ Sb | 2 (1) | 1 (2) | 22 | 32 | 0.742 | 1 | 0.389 |
| **Exp. 4** | | | ♀ Sb x ♂ Sb | ♀ Sb x ♂ Sh | ♂ Sb | ♂ Sh | **3.246** | **4** | **0.662** |
| 4 | 1 | ♀ Sb | 15 (12) | 17 (20) | 4 | 14 | 1.070 | 1 | 0.301 |
| 4 | 2 | ♀ Sb | 10 (8) | 25 (28) | 2 | 19 | 1.061 | 1 | 0.303 |
| 4 | 3 | ♀ Sb | 3 (3) | 8 (8) | 4 | 13 | 0.030 | 1 | 0.862 |
| 4 | 4 | ♀ Sb | 9 (8) | 15 (16) | 8 | 19 | 0.188 | 1 | 0.665 |
| 4 | 5 | ♀ Sb | 49 (49) | 15 (15) | 18 | 5 | 0.007 | 1 | 0.932 |
| **All Exp.** | | | | | | | **21.719** | **16** | **0.152** |

were once again not significantly different from expectations under a random mating hypothesis ($\chi^2$ = 4.522; d.f. = 1; P-value = 0.104, Table 3). Finally, regarding *S. bovis* females' choice (Table 2, Exp. 4) similarly we did not find a significant difference between the numbers of observed and expected homo-specific pairs and hetero-specific pairs under random mating hypothesis ($\chi^2$ = 3.246; d.f. = 4; P-value = 0.662, Table 2). Overall, when analysing all limited choice experiments together (*i.e.*, Exp. 1 to 4) no significant difference was recorded between the number of observed homo- and hetero-specific pairs and those expected under a random mating scenario ($\chi^2$ = 21.71, d.f. = 16, P-value = 0.152, Table 2).

**Table 3. Summarized information of experiment 5 (full choice).** For each combination (*i.e.*, sex and species) are given the number of observed pairs and the number of single partners that remained single. Sh = *S. haematobium* and Sb = *S. bovis*. Expected number of pairs under random mating is shown in brackets (see the statistics section in Materials and Methods for details). Chi squared statistics, degree of freedom and P-value are given per hamster and for the whole experiment.

| Host no. | | ♂*Sh* x ♀ Sh | ♂Sb x ♀ Sb | ♂Sh x ♀ Sb | ♂Sb x ♀ Sh | ♂ Sh | ♂ Sb | ♀ Sh | ♀ Sb | $\chi^2$-statistic | d.f. | P-value |
|---|---|---|---|---|---|---|---|---|---|---|---|---|
| | 1 | 1 (2) | (24) | 5 (9) | 4 (5) | 8 | 5 | 4 | 9 | 3.358 | 3 | 0.340 |
| | 2 | 8 (8) | 2 (1) | 5 (4) | 1 (2) | 8 | 3 | 23 | 8 | 1.786 | 3 | 0.618 |
| | 3 | 7 (5) | 2 (2) | 1 (2) | 2 (3) | 6 | 5 | 19 | 11 | 2.307 | 3 | 0.511 |
| | 4 | 4 (4) | 6 (3) | 3 (4) | 1 (3) | 10 | 4 | 13 | 11 | 4.806 | 3 | 0.187 |
| | 5 | 5 (6) | 1 (1) | 6 (6) | 2 (1) | 20 | 3 | 17 | 16 | 0.796 | 3 | 0.850 |
| **Total** | | | | | | | | | | **13.053** | **12** | **0.365** |

**Full choice: Experiment 5.**   Details on the number of worms recovered from each hamster and whether they were paired or single are summarized in Table 3. When all mating combinations were allowed between *S. haematobium* and *S. bovis*, four types of pairing combination were obtained: two being homo-specific (♂ Sh x ♀ Sh and ♂ Sb x ♀ Sb, Table 3) and two being hetero-specific (♂ Sh x ♀ Sb and ♂ Sb x ♀ Sh, Table 3). There was also an excess of males and females of both species remaining single, suggesting that all possible pairings were not limited by partner availability (Table 3). Regarding the number of homo-specific and hetero-specific pairs observed between *S. haematobium* and *S. bovis*, Chi-square tests did not reveal significant departure from random mating hypothesis, when the number of each pairing combination was analysed in each hamster separately and also when analysing all replicate together (Table 3).

## Transcriptomic response in homo- vs. hetero-specific pairs

**RNA sequencing, transcriptome assembly and gene annotation of the homo- and hetero-specific pairs.**   We have separately analysed 24 samples, corresponding to biological triplicates of males and females of the four forced pairing combinations described in Table 1 and Fig 2 (*i.e.*, homo- and hetero-specifically paired males and females). Between ~24.7 and ~42.3 million high quality Illumina HiSeq 4000 PE100 RNA-seq reads were obtained after sequencing of the 24 samples. After quality control and adaptor trimming, between ~19.2 and ~33,1 million reads were uniquely mapped to the *S. haematobium* reference genome and used for gene expression analysis [62]. On average ~78% of raw reads were mapped to the reference genome, with 51% of which corresponded to *S. haematobium* and 49% to *S. bovis* (S1 Table).

The reference transcriptome assembly on which tests were carried out, was composed of 73,171 putative isoform sequences identified as TCONS, and 18,648 unique genes identified as XLOCS. We conserved the longest isoform (TCONS) for each gene (XLOC) for subsequent annotation. The GTF and Fasta file of this transcriptome are available in a Figshare repository (https://doi.org/10.6084/m9.figshare.12581156, [68]). Blast annotations and Gene Ontology terms of the complete reference transcriptome are available in Sheet A S1 File. On the 18,648 genes, 14,414 found at least one hit following Blastx analysis, and 12,332 of them were mapped to at least one GO term using Blast2GO [67].

**Differential gene expression.**   Quantification of read abundance as well as differential gene expression analysis were performed on the 18,648 genes for each homo- and hetero-specific conditions (Sheet B, Sheet C, Sheet D, Sheet E and Sheet G in S1 File). The heatmap of the sample-to-sample distances as well as the principal component analysis plot are presented in Fig 3. A total of 1,277 genes (~7% of the 18,648 genes present in the reference transcriptome) were differentially expressed in at least one of the four homo- versus hetero-specific comparisons with a FDR <5% (Sheet G in S1 File). Of these, 1,234 (97%) had a match using Blastx against the non-redundant database of the NCBI and 1,088 (85%) were mapped and successfully annotated with at least one GO term using Blast2GO [67] (Sheet G in S1 File).

Most of the differentially expressed genes (DEGs) were identified in *S. haematobium* males, with 1,166 DEGs between the hetero-specific and homo-specific pairing combinations (734 over-expressed and 432 under-expressed in hetero-specific paired males compared to homo-specific ones). Log2-Fold changes were quite low with only one of these 1,166 DEGs having a Log2-Fold Change higher than 1.5 and none had Log2-Fold change lower than -1.5 (Fig 4, Sheet C and Sheet G in S1 File). In *S. haematobium* females, 47 genes were differentially expressed between hetero- vs. homo-specific conditions (22 over-expressed and 25 under-expressed in hetero-specific females). Among these 47 DEGs, six had Log2-Fold changes higher than 1.5 and one had a Log2-Fold change lower than -1.5 (Fig 4, Sheet D and Sheet G in

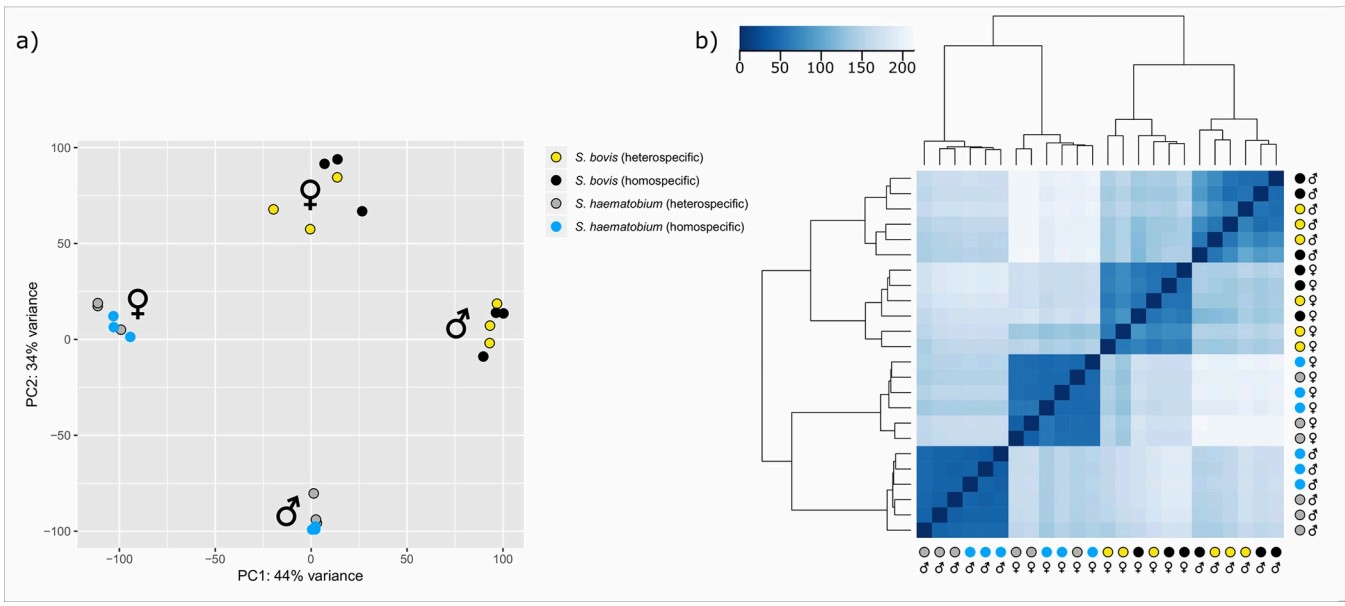

**Fig 3. Differential gene expression profiles.** a) Principal component plot of the samples and b) Heatmap of the sample-to-sample distances.

S1 File). In *S. bovis* females, 88 genes were differentially expressed between hetero- vs. homo-specific conditions (58 over-expressed and 30 under-expressed in hetero-specific females). Among these 88 DEGs, 48 had Log2-Fold changes higher than 1.5 and 11 had Log2-Fold changes lower than -1.5 (Fig 4, Sheet E and Sheet G in S1 File). Finally, no DEGs were identified in *S. bovis* males (Sheet F and Sheet G in S1 File). Significantly ($p<5\%$) over- and under-expressed genes (XLOC) for each comparison as well as their annotation are shown in Sheet G in S1 File.

**Gene Ontology and enrichment analysis of the differentially expressed genes.** Gene ontology categories significantly enriched in either over- or under-expressed genes were found in *S. haematobium* males (Fig 5, Sheet H in S1 File) whereas in *S. haematobium* females, *S. bovis* males and females (in which fewer DEG were detected), no GO terms were significantly enriched.

In *S. haematobium* males, biological processes enriched in under-expressed genes (in hetero-specific paired males compared to homo-specific ones) were related to signal transduction, notably through neuronal processes (synaptic transmission, cholinergic, chemical synaptic transmission, postsynaptic, G protein–coupled receptor signalling pathway), development (anatomical structure development), metabolism (glycogen biosynthetic process, negative regulation of endopeptidase activity), transmembrane transport (potassium ion transmembrane transport), response to stimuli (response to drug, peptidyl–proline hydroxylation, cell redox homeostasis) and cell adhesion (homophilic cell adhesion via plasma membrane adhesion molecules) (Fig 5, Sheet H in S1 File).

On the other hand, biological processes enriched in over-expressed genes (in hetero-specific males) were related to signal transduction including again some neuronal processes (*e.g.*, transmembrane receptor protein tyrosine kinase signalling pathway, regulation of Ras protein signal transduction, regulation of axon extension), metabolism (*e.g.*, proteolysis involved in cellular protein catabolic process, phosphatidylcholine metabolic process, long–chain fatty acid metabolic process, lipid droplet organization), response to stimuli (*e.g.*, response to other

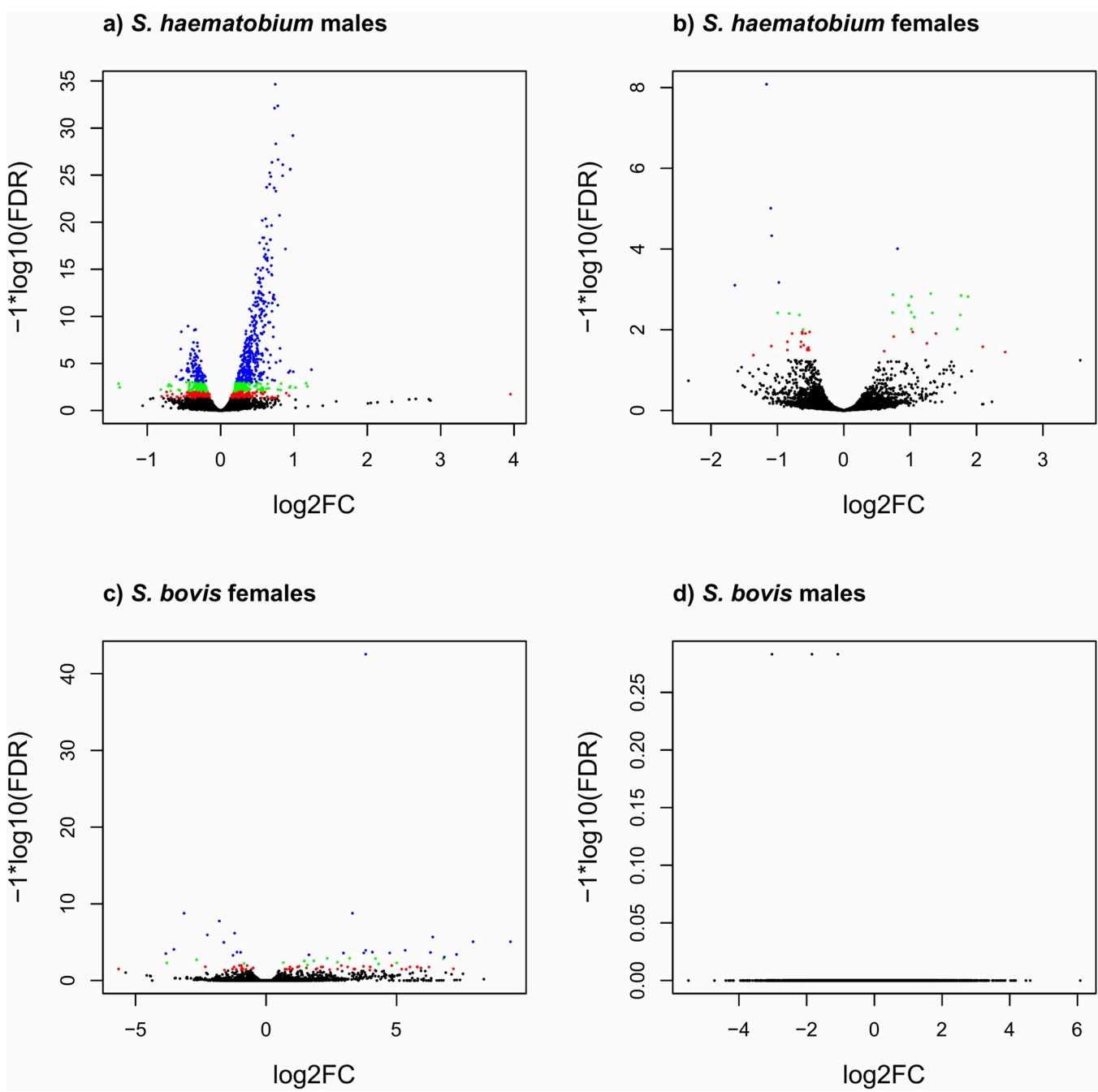

**Fig 4. Genes expression profiles in hetero-specifically compared to homo-specifically paired worms.** Volcano plots showing the log transformed adjusted P-values (*i.e.*, FDR) and the log fold changes for the 18,648 unique genes of the reference transcriptome assembly for *S. haematobium* males *a*), *S. haematobium* females b), *S. bovis* females c) and *S. bovis* males d). Black dots refer to non-significant genes regarding their expression profile (over an FDR of 5%). Red dots refer to differentially expressed genes at a FDR of 5%, green dots refer to differentially expressed genes at a FDR between 5% and 1% and blue dots refer to DEGs at a FDR between 1% and 1 ‰.

organism, phagocytosis, cellular response to chemical stimulus), transmembrane transport (*e. g.*, anion transmembrane transport, vesicle fusion, regulation of vesicle−mediated transport, inorganic cation import across plasma membrane, exocytosis, positive regulation of Notch signaling pathway), localization (*e.g.*, establishment of localization in cell), locomotion (*e.g.*,

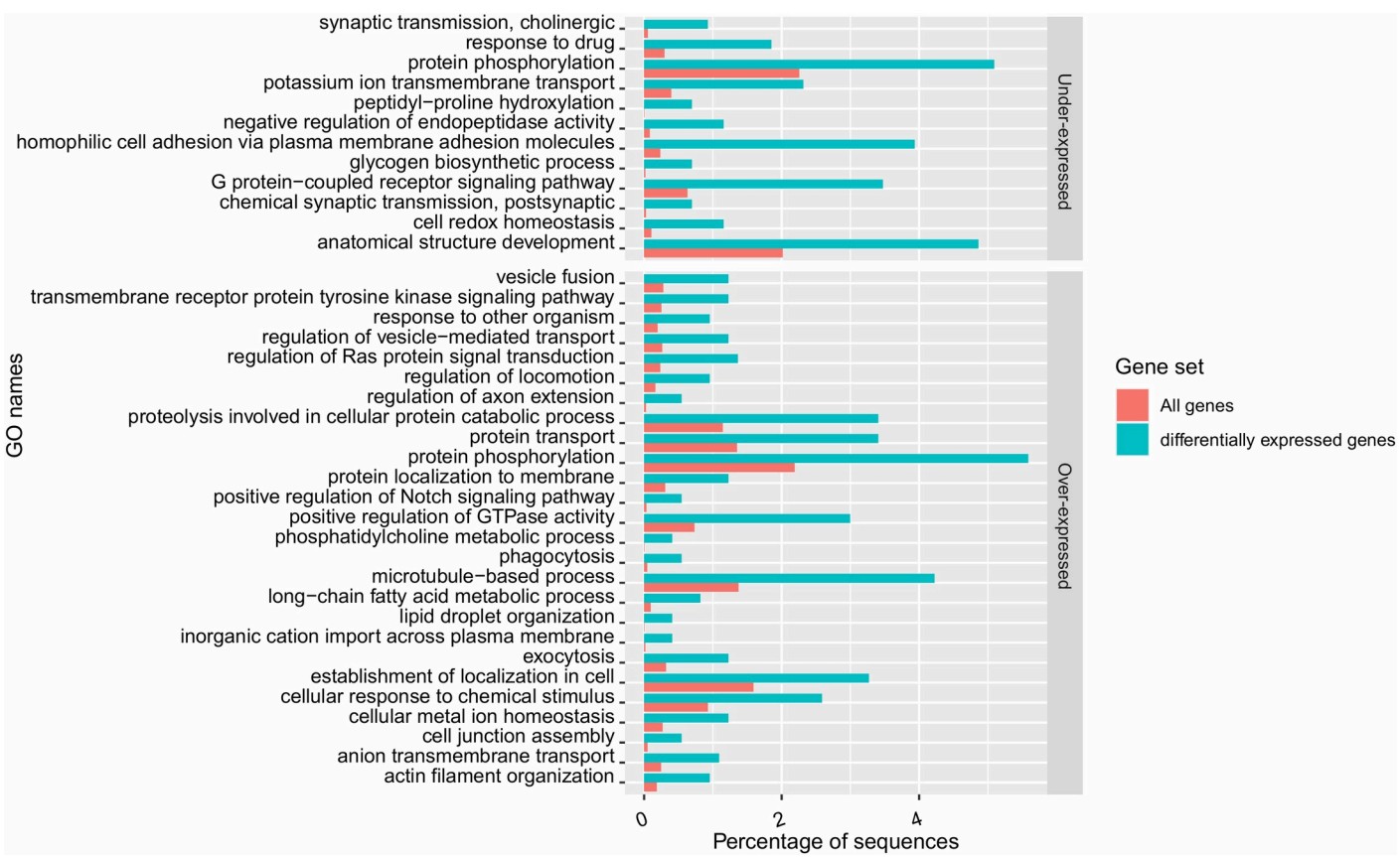

**Fig 5. Biological processes impacted by hetero-specific pairing in male *S. haematobium*.** Barplot showing the biological processes significantly enriched in DEGs (at a FDR threshold of 5%), either over-expressed or under-expressed in hetero-specific condition compared to homo-specific condition, in *S. haematobium* males.

regulation of locomotion, microtubule–based process, actin filament organization) and also cell adhesion (*e.g.*, cell junction assembly) (Fig 5, Sheet H in S1 File).

No GO terms were found enriched neither in over- nor under-expressed genes in *S. bovis* and *S. haematobium* hetero- vs. homo-specifically paired females. However, based on annotations, in *S. haematobium* females, we found differentially expressed genes that corresponded to genetic mobile elements (*e.g.*, XLOC_014282: integrase core domain, XLOC_014741: TPA: endonuclease-reverse transcriptase, XLOC_009783: endonuclease-reverse transcriptase), genes involved in transmembrane transport (*e.g.*, XLOC_009318: phosphatase methylesterase 1 (S33 family) and XLOC_010891: Calcium-binding mitochondrial carrier S -1), stress response including oxidation-reduction processes (*e.g.*, XLOC_017856: heat shock, XLOC_012518: epidermal retil dehydrogese 2 and XLOC_018492: iron-dependent peroxidase) and other functions such as reproduction, or development (*e.g.*, XLOC_015776: egg CP391S, XLOC_007823: Craniofacial development 2) (Sheet G in S1 File). Similarly, in *S. bovis* females, we found differentially expressed genes that correspond to genetic mobile elements as well (*e. g.*, XLOC_017328: R-directed D polymerase from transposon X-element, XLOC_018050: R-directed D polymerase from mobile element jockey-like or XLOC_018156: gag-pol poly), genes involved in ion transport (*e.g.*, XLOC_008268: Bile salt export pump, XLOC_003851: sodium-coupled neutral amino acid transporter 9 isoform X2 and XLOC_005754: Y+L amino acid transporter), response to stress (*e.g.*, XLOC_017856: heat shock and XLOC_009339: Universal stress) as well as other functions such as reproduction, growth or metabolism (*e.g.*,

XLOC_014939: early growth response, XLOC_015393: Syptotagmin-1, XLOC_016728: egg CP391S-like and XLOC_012591: Cathepsin B-like cysteine proteinase precursor) (Sheet G in S1 File). Hence, for *S. haematobium* and *S. bovis* females, DEGs were quite similar in term of function, regardless of their expression profile (under- or over-expression in hetero-specific pairs) and regardless of the schistosome species.

## Discussion

In this study we aimed to investigate potential reproductive isolation mechanisms between two major African schistosome species that cause major debilitating parasitic disease and show evidence of extensive hybridization in nature [37,69]. Specifically, we tested whether hybridization between *S. haematobium* and *S. bovis* could be constrained or promoted by mate choices and whether these mate choices were associated with specific transcriptomic profiles in hetero- and homo-specifically paired individuals. Overall, the data shows that *S. haematobium* and *S. bovis* mate in a random fashion and depend only on the presence and the relative abundance of each species in the definitive host. Likewise, we did not detect any major transcriptomic changes associated with hetero-specific pairing in male and female *S. haematobium* and *S. bovis*.

First, we showed that the two frequently co-endemic sister species *S. haematobium* and *S. bovis* readily pair with no preferences for neither homo-specific nor hetero-specific associations in simultaneous infections. The only exception was found for male *S. haematobium* mate choice. Indeed, we found a significantly higher number of hetero-specific pairs compared to that expected under the assumption of random mating. However, as we cannot differentiate mate recognition initiated by males from female competition, our results suggest that either male *S. haematobium* prefer mating with female *S. bovis* or alternatively, that female *S. bovis* may be more competitive than female *S. haematobium*. Interestingly, although female competition is possible, it is assumed that male schistosomes are the competitive sex and in particular male *S. haematobium* are usually better at pairing when compared to males from other species including *S. intercalatum* (now *S. guineensis)* [35], *S. mattheei* [29] or *S. mansoni* [28]. However, since the bias toward hetero-specific pairing was observed in a unique hamster, this result should be considered with caution. Indeed, this bias was not retrieved in our full mate choice experiments and future studies are warranted to confirm if this observation is repeatable as it may have important epidemiological consequences regarding pairing directionality and hybrid representation in the field. Similarly, premature death of some hamsters in the experiment focusing on the mate choice of male *S. bovis* limited our ability to draw specific conclusions. Consequently, our mate choice experiments overall rather indicate no differences in species mate choice or competitiveness and that *S. haematobium* and *S. bovis* males and females mate randomly. Such a result is in line with Webster and colleagues [27]. Altogether, this highlights that there are no behavioural barriers preventing hetero-specific pairing once both species encounter each other in the same definitive host.

The second part of this study aimed to assess the transcriptomic profiles associated with hetero-specific pairings between *S haematobum* and *S. bovis*. Since different species might constitute a different stimulus for the other partner, we expected at first to find an impact of the hetero-specific pairing, and especially on female transcriptomes compared to male transcriptomes since they respond to male stimuli for their sexual maturation [20]. However, only few DEGs were observed in both males and females. Biological processes enriched in DEGs were identified only for male *S. haematobium* pairings. Likewise, most of the genes detected presented low Log2-Fold changes (notably in *S. haematobium* males where only one DEG exceeded a Log2-Fold change of 1.5). Thus, the influence of hetero-specific pairing on male

and female adult worms of both species in terms of numbers of DEGs, related biological processes and gene expression level was not striking. Such results suggest that both species may be highly receptive to each other since no major transcriptomic adjustments are induced by hetero-specific pairings. This observation is hence consistent with our previous mating experiments that suggest random pairing between both species and further show that there are no major physiological nor molecular barriers making hetero-specific pairings and thus hybridization less prone to occur.

Although hetero-specific pairings did not result in many DEGs, it is worth noting that most of the DEGs were found in the comparison between homo- and hetero-specifically paired male *S. haematobium*. So far transcriptomic studies on *Schistosoma* pairing tended to show large molecular reprograming of female genes rather than male genes, in part due to the initiation of their sexual maturation [13,47,48]. The biological explanations for our results are thus not straightforward. First, we cannot rule out the possibility of an artefact induced by extrinsic factors or other technical issues such as a lower variability in the transcriptomic profiles of the different biological replicates of male *S. haematobium* in comparison to other samples. However, our results also show that male *S. haematobium* displayed more DEGs than females but DEG identified in females presented overall higher log2 Fold Changes. Hence, another hypothesis could be that females may differentially express fewer genes, but at higher levels. Finally, we could also hypothesize that the molecular plasticity in expression of genes is a mechanism by which male *S. haematobium* manage to be more competitive (compared to females from both species and *S. bovis* males) in hetero-specific pairing, for instance by properly initiating female maturation depending on their species. This latter hypothesis is particularly appealing since male *S. haematobium* are thought to be dominant over several other *Schistosoma* species [28,29,35]. This is also congruent with the potential bias toward hetero-specific pairing of male *S. haematobium* found in our mating experiments and also with field studies that show that the majority of the hybrids in the field appear to be a result of a cross between male *S. haematobium* and female *S. bovis* [37,70]. Nevertheless, since we did not identify any DEG in *S. bovis* males, and also because the log2-Fold change of the DEG identified in *S. haematobium* males were low, it seems difficult to conclude that one or the other sex is preferentially impacted during hetero-specific pairing, or that one species is more prone to initiate the sexual maturation of females. However, we are confident that the small number of DEGs identified when comparing homo- and hetero-specific parings together with their low log2 Fold Change reflect the relatedness between *S. bovis* and *S. haematobium* that undergo only few transcriptomic adjustments following hetero-specific pairing. Moreover, the molecular changes that we identified here at the very first step in the hybridization process may reveal some important genes linked to male and female interactions, species isolation and hybridization.

Indeed, some of the DEGs identified in our work show functions that can be linked to sexual interactions, notably to reproductive functions suggested by other studies. Notably, among female schistosomes we found three genes encoding egg proteins that were differentially expressed in *S. haematobium* and/or *S. bovis* females, and that are well-known female-associated gene products [71]. Similarly, a transcript matching the Syptotagmin-1 gene was under-expressed in hetero-specifically paired female *S. bovis*. This gene was previously shown to have a female-specific expression and to be regulated during pairing [47]. Moreover, two DEGs that encode digestive enzymes, specifically expressed by paired females (*i.e.*, cathepsin B and L) were found in female *S. bovis* [71]. Similarly, in *S. haematobium* DEGs were related to biological processes known to be involved in male-female interactions. Previous studies looking at the molecular basis of *Schistosoma* male-female interaction, with a particular interest in the pairing process, proliferation, differentiation and maturation of female gonads, have

underlined the major role of signal transduction cascades and particularly signalling pathways such as the TGF-beta and Ras (*e.g.*, receptor tyrosine kinase coupled pathway) signalling pathways [72–78]. These pathways, notably the TGF-beta signaling pathway are known to induce the production of the gynecophoric canal protein by males during pairing which is a trigger for maturation of females [73]. Interestingly, in this work, among genes whose expression was affected by homo- and hetero-specific pairing in male *S. haematobium*, we notably found the TGF-beta signal transducer gene, and two gynecophoral canal protein genes. Moreover, both transmembrane receptor protein tyrosine kinase signaling pathway and regulation of Ras protein signal transduction processes were enriched in over-expressed genes in hetero-specific pairs. Also echoing more recent studies on the gonad-specific and pairing-dependent transcriptomes of male schistosomes, we found several biological processes enriched either in over- or under-expressed genes in *S. haematobium* males that were involved in neuronal processes which are associated with male-female interaction patterns [13,48]. We consequently found that genes and processes impacted between homo- and hetero-specific pairing in *S. haematobium* and *S. bovis* at least partly overlapped those generally affected in other male-female interaction studies. These results suggest that both species may have maintained similar patterns of interactions between males and females allowing them to reproduce. A moderate regulation of these genes during pairing with another species may thus allow the two parasite species to overcome their divergence resulting in successful hetero-specific mating. Finally, it is worth noting that among the DEGs identified, the majority of them were also related to processes that were not particularly documented to be impacted during male-female interactions (*e.g.*, genetic mobile elements, response to drug and stimuli, oxidation-reduction). Several DEGs were related to stress response and stimuli responses (*e.g.*, oxidation-reduction processes as well as the genetic mobile elements [79,80]), indicating that at least at the molecular level schistosome species may perceive hetero-specific pairing as a stress, although this does not seem to impede hetero-specific pairing. Alternatively, the pairing status (*i.e.*, homo-specific or hetero-specific) could impact the worms' responses to external stimuli including host and/ or environmental stimuli. In particular hetero-specific male *S. haematobium* under-expressed a fair amount of genes involved in response to drugs compared to homo-specific ones (*e.g.*, Multidrug and toxin extrusion, Multidrug and toxin extrusion 2, Multidrug resistance or Multidrug resistance-associated). These observations may raise important questions regarding schistosomes' drug response in the context of co-infection and hybridization especially since a lower sensitivity to PZQ of *S. bovis* x *S. haematobium* hybrids compared to pure *S. haematobium* parasites has been proposed to be at the origin of the spread of the hybrid form in Senegal [27]. However, it is important to pinpoint that any changes associated to PZQ response in hybrids is still theoretical and there is not current evidence that there is any difference in drug response in natural infections. Here we found a differential expression of genes involved in response to drugs in male *S. haematobium* only, which call for future clarification to assess if this is a peculiarity of our study and/or of male *S. haematobium*. More generally, several other genes identified in this work may be of potential significance for the encounter, interaction, and communication between these two species. Further attention is thus required to decipher the role of each of them in the context of hybridization or at the contrary in the context of speciation.

Altogether, the integrative assessment of lack of pre-zygotic reproductive mechanisms we present here may have profound implications regarding what we could expect in term of hybridization dynamics in the field. In particular it suggests that both species have retained similar processes allowing them to find their partner in the host, pair and produce viable offspring. This result is in line with several recent studies that have presented evidences of introgression between *S. haematobium* and *S. bovis* and that suggest that their relatively recent

divergence compared to other schistosomes and thus the genetic distance between both species is not sufficient to limit hybridization [39,40,81]. This relies in part in the fact that they have retained the same karyotype with n = 8 chromosome pairs, including sex chromosomes that are morphologically similar [82], hence allowing the species' genomes to be highly permeable to each other's alleles [83]. In that case, the most significant reproductive isolation mechanisms preserving the genetic integrity between these species would be habitat isolation, including geographical location and definitive host specificity. Also, it is worth noting however that the two species used in this study have been isolated from distinct geographical zones and this could contribute to explain the absence of pre-zygotic isolation. Indeed, sympatric African schistosome species are likely to respond differentially as sympatric species tend to have enhanced pre-zygotic isolation barriers [4]. Nevertheless, the lack of pre-zygotic barriers does imply that in areas where *S. haematobium* and *S. bovis* are sympatric and infect the same definitive hosts, hybrids and introgressed individuals should be more likely to be found. This may be particularly relevant for parasite species that are brought together by global changes (enhanced human migration for *S. haematobium*, and animal transhumance for *S. bovis)* and may have porous reproductive isolation mechanisms.

While our study opens new avenues regarding the understanding of the mechanisms allowing or preventing hybridization between schistosome species, it also calls for future experimental and field work to fully understand hybridization patterns observed *in natura*. First, our observation of random mating between the two species suggests that first-generation hybrids may be frequent in endemic areas. A recent study in Senegal found hybrids with mixed genetic profiles between parental species suggesting that they may be of early generation [45]. However, current genomic analyses of parasites recovered in the field indicate that introgression between *S. haematobium* and *S. bovis* is the result of an ancient event rather than an ongoing process [40,81,84]. This is also supported by the genetic differentiation between hybrids and parental species populations in Senegal and Niger [85,86]. Second, although a broader view of the hybridization dynamics is warranted by increasing the number of samples collected across the African continent, the current data suggests that at least in the field *S. haematobium* could be dominant over *S. bovis* and that hybridization patterns may differ between foci. Indeed, several studies report unidirectional introgression of *S. bovis* genes into *S. haematobium* [36,40] and a predominance for an initial cross between a male *S. haematobium* and a female *S. bovis*, (leading to the introgression of mitochondrial DNA of the latter in the genomic background of the former [26,37]). Such biases in the direction of the crosses and introgression patterns are frequent in the hybridization landscape. For instance while some species hybridize in both directions and over multiple generations (*S. bovis* and *S. curassoni*; [27,36]; *S. mansoni* and *S. rodhaini*; [69]), others may produce offspring with strong asymmetries in their fitness (*S. haematobium* and *S. mattheei*;[87], *S. haematobium* and *S. intercalatum (*now *S. guineensis)* [29]) and sometimes in the directionality of introgression (*S. rodhaini* and *S. mansoni* [28]). However, since our analysis of the pre-zygotic isolation mechanisms does not support any type of asymmetry in the direction of the crosses it is most likely that if any, post-zygotic barriers may be at the origin of such biased patterns in the field and also potentially the relatively rare encounter in early generation hybrids. Consequently, the genomic landscape of introgression and the transmission patterns of hybrids may not be uniform, are highly complex and potentially dynamic. In this context it would be necessary as a next step to assess the importance of post-zygotic isolation mechanisms in terms of snail compatibility, hybrid life history traits and potential heterosis, which are important biological features that may shape hybridization outcomes by potentially reducing or promoting inter-species interaction and admixture. This may have strong implications as hybridization in schistosomes is a major concern and since heterosis in offspring may increase the parasite virulence compared to their parental species

[88,89]. Such changes in the parasites life history traits may have important outcomes in terms of epidemiological dynamics (hybrids may take over parental species range [90], but also threaten the transmission, control and ultimate elimination of schistosomiaisis). In this context, a better understanding of the consequences of hybridization in parasites is a necessary next step to anticipate its effect in terms of disease dynamics and spread.

In conclusion, in this integrative study of *S. haematobium* and *S. bovis* behavioural and physiological isolation mechanisms we showed that natural hybridization between *S. haematobium* and *S. bovis* lack strong pre-zygotic barriers apart from their host specificity. Our data suggest that no mate recognition system mitigates hybridization between these two species and that no major transcriptomic adjustments are associated with hetero-specific pairings. This highlights that the two species remain sufficiently coadapted to each other to allow an efficient reproduction once they are in contact. Besides the current evidence of ancient introgression and biases in hybrid profiles, this weak pre-zygotic isolation exemplified raises the risk that in the absence of other reproduction isolation mechanisms, hybridization between these two species may be common. This also implies that contact zones may need further consideration to assess if hybridization is ongoing. Finally, our results may also partly explain the high prevalence of these hybrids in the field. Because such inter-species interaction may increase the offspring's virulence compared to parental species, one could expect to find increased prevalence and intensities of the disease in areas where hybridization occurs. Understanding the modifications in the parasite life history traits, including their zoonotic potential and epidemiological outcomes are warranted to control human and animal morbidity, reduce transmission and ultimately eliminate schistosomiasis.

## Supporting information

**S1 Table. Metrics of the RNA-sequencing reads processing**
(XLSX)

**S1 File.** Transcriptome analysis related information: **Sheet A in S1 File**: Table showing the full annotation of the reference transcriptome representative of *S. haematobium* and *S. bovis* assembled for this study. **Sheet B in S1 File:** Table of the transcript counts (*i.e.*, HTseq) in each triplicate of each condition. **Sheet C in S1 File:** DESeq2 results for male *S. haematobium*. **Sheet D in S1 File:** DESeq2 results for female *S. haematobium*. **Sheet E in S1 File:** DESeq2 results for female *S. bovis*. **Sheet F in S1 File:** DESeq2 results for male *S. bovis*. **Sheet G in S1 File:** Annotations associated to each differentially expressed genes that have been identified. **Sheet H in S1 File**: Gene Ontologies enriched in differentially expressed genes in male *S. haematobium*.
(XLSX)

## Acknowledgments

We would like to acknowledge Génome Québec and University McGill innovation center as well as Roscoff Bioinformatics platform ABiMS (http://abims.sb-roscoff.fr) and NGS facility at the bio-environment platform (University of Perpignan). This study is set within the framework of the "Laboratoires d'Excellence (LABEX)" TULIP (ANR-10-LABX-41).

## Author Contributions

**Conceptualization:** Eve Toulza, Jérôme Boissier.

**Data curation:** Cristian Chaparro, Jean-Francois Allienne.

**Formal analysis:** Julien Kincaid-Smith, Eglantine Mathieu-Bégné, Marta Reguera-Gomez, Stephen Mulero.

**Funding acquisition:** Eve Toulza, Jérôme Boissier.

**Investigation:** Julien Kincaid-Smith, Eglantine Mathieu-Bégné.

**Methodology:** Julien Kincaid-Smith, Cristian Chaparro, Eve Toulza, Jérôme Boissier.

**Supervision:** Eve Toulza, Jérôme Boissier.

**Visualization:** Julien Kincaid-Smith, Eglantine Mathieu-Bégné.

**Writing – original draft:** Julien Kincaid-Smith, Eglantine Mathieu-Bégné, Eve Toulza, Jérôme Boissier.

**Writing – review & editing:** Julien Kincaid-Smith, Eglantine Mathieu-Bégné, Cristian Chaparro, Eve Toulza, Jérôme Boissier.

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
