## [Decision Letter · Decision Letter 0]

13 Oct 2020

Dear Mrs Mathieu-Bégné,

Thank you very much for submitting your manuscript "Pre-zygotic isolation mechanisms
between Schistosoma haematobium and Schistosoma bovis parasites: from mating
interactions to differential gene expression" for consideration at PLOS Neglected
Tropical Diseases. As with all papers reviewed by the journal, your manuscript was
reviewed by members of the editorial board and by several independent reviewers. In
light of the reviews (below this email), we would like to invite the resubmission of
a significantly-revised version that takes into account the reviewers' comments. 

All three reviewers have commented on the level of English and grammar within this
paper, that makes it hard to properly review this paper. I would recommend that in
addition to all that scientific comments and specific language comments included
that all authors really help rewrite this manuscript and if needed that you ask a
native English speaker to help with the next version. I will send this out to
reviewers again upon resubmission if the level of English has improved, but that
really does need to be addressed.

We cannot make any decision about publication until we have seen the revised
manuscript and your response to the reviewers' comments. Your revised manuscript is
also likely to be sent to reviewers for further evaluation.

Sincerely,

Poppy Lamberton

Deputy Editor

All three reviewers have commented on the level of English and grammar within this
paper, that makes it hard to properly review this paper. I would recommend that in
addition to all that scientific comments and specific language comments included
that all authors really help rewrite this manuscript and if needed that you ask a
native English speaker to help with the next version. I will send this out to
reviewers again upon resubmission if the level of English has improved, but that
really does need to be addressed.

Reviewer's Responses to Questions

**Key Review Criteria Required for Acceptance?**

**Methods**

-Are the objectives of the study clearly articulated with a clear testable hypothesis
stated?

-Is the study design appropriate to address the stated objectives?

-Is the population clearly described and appropriate for the hypothesis being
tested?

-Is the sample size sufficient to ensure adequate power to address the hypothesis
being tested?

-Were correct statistical analysis used to support conclusions?

-Are there concerns about ethical or regulatory requirements being met?

Reviewer #1: -Are the objectives of the study clearly articulated with a clear
testable hypothesis stated?

Yes

-Is the study design appropriate to address the stated objectives?

Yes but more discussion is needed on the limitations of the study design 

-Is the population clearly described and appropriate for the hypothesis being
tested?

More information is needed on the strains used and the details of the experiments 

-Is the sample size sufficient to ensure adequate power to address the hypothesis
being tested?

Yes 

-Were correct statistical analysis used to support conclusions?

I would like this checked by a statistician 

-Are there concerns about ethical or regulatory requirements being met?

No

Reviewer #2: - The overall objective is clear but the hypotheses could be more
clearly formulated in the introduction. For example, somewhere in the paper it was
stated that the authors expected the opposite outcome (more transcriptomic changes
in females than in males), so this hypothesis could have been included at the end of
the introduction, instead of the vague kind of conclusion in line 194-197.
Additional testable hypotheses can be formulated.

- The study design is appropriate although some issues arise regarding statistical
power. The results discussed in line 362 (premature death of two hamsters) does
raise the question whether 5 replica’s (5 hamsters) is enough to make robust
conclusions as in this particular case no reliable statistics can be done for S.
bovis males’ choice.

- No concerns about ethical issues.

For more comments on the Methods section see General Comments.

Reviewer #3: Methods seem appropriate

**Results**

-Does the analysis presented match the analysis plan?

-Are the results clearly and completely presented?

-Are the figures (Tables, Images) of sufficient quality for clarity?

Reviewer #1: -Does the analysis presented match the analysis plan?

Yes 

-Are the results clearly and completely presented?

Improvement is needed for clarity of the results to allow interpretation 

-Are the figures (Tables, Images) of sufficient quality for clarity?

More information and clarity is needed as detailed in the attached document

Reviewer #2: The analysis is sound and the results are well presented. 

Table 2 is redundant I would say.

For more comments on the Results section see General Comments.

Reviewer #3: Analysis matches described goals of the work.

REsults are clearly presented.

Figures are sufficient quality

**Conclusions**

-Are the conclusions supported by the data presented?

-Are the limitations of analysis clearly described?

-Do the authors discuss how these data can be helpful to advance our understanding of
the topic under study?

-Is public health relevance addressed?

Reviewer #1: -Are the conclusions supported by the data presented?

Yes 

-Are the limitations of analysis clearly described?

No

-Do the authors discuss how these data can be helpful to advance our understanding of
the topic under study?

Ye

-Is public health relevance addressed?

Yes

Further comments on the discussion are available in the attached

Reviewer #2: As put in my General comments, the Discussion needs reworking and should
be more substantiated. The conclusions are not always clear or strong, I miss more
references to similar studies, but also a proper discussion on the limitations and
recommendations for future research are missing. The public health relevance is not
really thoroughly discussed.

Reviewer #3: Conclusions are supported by the data. They speculate quite a bit in the
discussion about the potential importance of various DE genes. But this type of
speculation is rampant in gene expression papers and they don't make any hard
conclusions.

**Editorial and Data Presentation Modifications?**

Reviewer #1: The main modidications are needed in the clarity of the data and how it
is presented. Comments are in the attached document.

Reviewer #2: (No Response)

Reviewer #3: (No Response)

**Summary and General Comments**

Reviewer #1: This is an important and interesting study that shows the lack of pre
zygotic isolation between S. haematobium and S. bovis supporting inter-species
hyrbidisation an important scenario for human and animal health in Africa. Although,
I consider the study of vlaue and there is a substantial body of high quality of
work, which is not easy to do (particularly the generation of the isolates and the
mating experiments) the paper needed considerable improvements before it can be
reviewed further. The authors should consider that the readers will not be familiar
with these types of experiments and there is a need to add the detail that allows
the readers to understand the experimental procedures and the data produced. There
are also english and gramma errors many of which can be improved with careful
reading. In the attached these are highlighted in the text showing the errors and
where changes are needed. I have tried to cover the whole document but due to time
some places may have been missed and careful reading and revision is needed. There
are also comments on the attached to highlight where more clarity is needed and also
where further information or discuss is warranted. The discussion is also very long
and could be condensed by not repeating what is in the results.

Reviewer #2: This is a very interesting study, with interesting results. Experimental
infections in hamsters show that there are no pre-zygotic barriers to mating between
the human Schistosoma haematobium and the animal S. bovis parasite, neither are
there any major transcriptomic responses following hetero-specific pairings. Even
though a previous study by Webster and colleagues already showed that S. bovis and
S. haematobium readily paired in laboratory hamsters (to which the present authors
not refer, which I think is an omission), this is the first time that any
transcriptomic study is done on hybrid crosses. 

I was frustrated by the sloppy grammar and spelling throughout the entire manuscript,
which gave the impression that the authors were in a hurry to submit this manuscript
or they didn’t really care about this. 

I also lack a discussion on the asymmetry in the direction of hybridisation and
introgression between schistosomes, which is sometimes even unidirectional, but
nothing is mentioned on this. Also, it is repeatedly said that isolation ‘by the
host’ is apparently the only barrier to hybridisation between these two schistosome
species, but it is never specified which host they mean with that, the final or the
intermediate one. Since mate choice and reproduction takes place in the final host,
this one is of particular interest of course, but if the different intermediate
hosts are not sympatric, then hybridisation will also be less frequent (if they
final host does not move around too much). For S. bovis and S. haematobium this is
more complicated, as intermediate host specificity of the latter varies with
geography, but still this should be properly discussed as it could also explain why
we see such regional differences in the distribution of hybrids. Then finally, I
also have some problems with parts of the discussion: some questions remain
unanswered (e.g. why would only S. haematobium males in heterospecific pairing have
this transcriptomic response?), the statistical power, and with the fact that I miss
the broader picture, the reference to all the previous work on schistosome mating
experiments. So therefore I think that these concerns should be addressed first
before it can be accepted for publication.

Abstract & author summary

- line 44: make two sentences out of this long sentence

- line 47: delete ‘allowing them to maintain…’ this is repetition from above

- line 48: what do you mean with misunderstood? Not understood? Or are there really
mistakes and / or misconceptions out there in the literature?

- line 57: ‘by the host’: replace with ‘final host choice’ or something like that,
because at the snail host level there is not always spatial isolation

- line 67-68: evolutionary biology?

- Line 68: replace ‘If’ by ‘While’

- Line 73: S. haematobium (species names always in full when first mentioning) and
parasitize

- Line 70-74: sentence too long, split in two and rephrase ‘including out of endemic
areas’

- Line 75: ‘…rather than having a homo-specific mate preference’

- Line 78: mechanisms 

- Line 78: ‘but the one imposed by host specificity’? what do you mean? Please
rephrase ‘except the one imposed by final host specificity’

- Line 81: ‘encounter each other’

Especially the author summary does not read very fluently, and there are quite some
grammar mistakes.

Introduction

- line 88= mechanism

- line 90: this sounds more like mate choice rather than behavioural isolation

- line 91: individuals

- line 91: I would replace copulation by reproduction

- line 93: encounter

- line 96: less fertile

- line 96: hybrid lines

- line 98: making it difficult to predict

- line 104: terms 

- line 105 and others: free-living 

- line 108: compared to those of free-living…

- line 109: hostile rather than inimical?

- Line 112-113: rephrase this sentence, it is not really shaped through the host
alone, it is shaped through the host – parasite interactions, and you first call
this a strong isolation mechanism and a few words later you say ‘potentially’
preventing hybridization… the sounds less convinced. Also, you write ‘its hosts’, so
plural, I would make the distinction already here between intermediate and final
hosts, because in case of schistosomes sexual reproduction only takes place in the
final host, so host choice at this level is more important than at the other level.
You should discuss it at least somewhere, because in the abstract you only talk
about ‘host’ choice.

- Line 113: you mean ‘closely related’ species? Because all species are related
somehow…

- Line 116: plasmodium species (or you provide the genus names for the other two
parasites you add here)

- Line 123: rephrase ‘schistosomiasis debilitating diseases’, this is not an official
term (also, the disease is of great concern, rather than the parasites themselves I
would say)

- Line 128: rephrase ‘among other trematodes’, you want to say here that they are an
exception within the Trematoda

- Line 133: ‘this can lead to hybridisation’

- Line 136: influence rather than interest

- Line 136: First,

- Line 138: same host individual and schistosome species

- Line 144: female’s

- Line 145: of a sexually …

- Line 145: what does not depend upon species-specific pairing? Discuss this in a
separate sentence in order to avoid too long sentences

- Line 147: stimulate

- Line 149: male worms’ physiology or the physiology of male worms

- Line 153: groups instead of clades?

- Line 155: schistosome

- Line 157: others, you mean other combinations?

- Line 158: S. mattheei

- Line 156-159: in all these cases it should be mentioned that the viability of these
crosses depend on the type of crosses, which parental species provides the male and
which the female in the hybrid cross. Also, the way you write this you suggest that
the first group of species can readily pair because there is less divergence between
them (because this is what you write in the preceding sentence), while others are
more selective because they are more divergent… but the divergence between S.
haematobium and S. intercalatum is similar to the divergence between S. haematobium
and S. mattheei. Also how can a combination be more selective or readily pair? It is
the species that forms this combination that can be selective I would say. The
grammar is quite sloppy in many cases, please take care of this.

- Line 174: non-human; also Cetartiodactyla is a superorder, and a superorder or a
genus cannot be infected by parasites, but their members can

- Line 175: ruminants

- Line 177: rodents

- Line 191: outperforms the fitness of parental species

- Line 194: the sentence starting with ‘Relying on such an integrative…’ is redundant
as it is mainly repetition

Material and methods

- Line 214: in or with Bulinus

- Line 219: B. truncatus

- Line 223: were performed

- Line 242: versus

- Line 244: each worm and its

- Figure 1: I am a bit confused why you use the same color red for S. haematobium and
S. bovis females?

- Table 1: line 248: ‘and’ should not be in italic. Line 250: the number of male and
female S. haematobium and S. bovis worms

- Line 254: rephrase this sentence, grammatically incorrect

- Table 2: I think this table can be left out as everything is already explained in
the text

- Line 278-279: parasite species names not in italic since the title is in italic

- Line 289: rephrase: as the volume of each reagent was halved

- Line 296: library construction

- Line 302: vs in full and not italic

- Line 307: all sample reads

- Line 318: each newly assembled gene or all newly assembled genes

- Line 319: were extracted from the S. haematobium

- Line 321: transcript

- Line 324: hetero-specific paired worms (or elsewhere you write hetero-specifically
paired worms)

- Line 327: gene expression

- Line 338: sets of genes

Results

- line 349: experiment

- line 353: partners

- line 362: these are the risks of experimental research of course, and this cannot
be avoided, but it does raise the question whether 5 replica’s (5 hamsters) is
enough to make robust conclusions as in this particular case no reliable statistics
can be done for S. bovis males’ choice

- line 367: find instead of found

- Table 3, line 376: remaining single or that remained single

- Line 379: elsewhere you write P-value

- Line 380: experiments

- line 385: male and female S. haematobium

- line 412: providing from?

- Line 426: does this 7% means 7% of all schistosome genes? Please specify this 

- line 427: vs. in full as elsewhere

- line 462: GO terms 

- line 463: male S. haematobium

- line 468: biological

- line 489: and not in italic

- line 500: correspond (in present tense)

- line 503: involved in ion

- line 506: functions

Discussion

- line 513: reproductive isolation mechanisms?

- Line 521: male and female

- Line 523: definitive host. This sentence is actually a final conclusion before
starting the Discussion itself

- Line 532: S. mattheei

- Line 534: so this calls for more replica’s in future experiments to verify this
possibility!

- Line 539: there are no barriers or there is no barrier

- Line 557: previous instead of precedent

- Line 558: there are no 

- Line 560: what do you mean with ‘male and female S. haematobium and S. bovis’?
Between male S. haematobium and female S. bovis?

- Line 561: male S. haematobium

- Line 571: display a more 

- Line 576: ‘females species sexual maturation’?

- Line 587: female S. bovis

- Line 602: gonad-specific

- Line 579 – 612: this lengthy discussion is confusing and less convincing because
right before this discussion you conclude that only few transcriptomic adjustments
are associated with hetero-specific pairing and that the log2-Fold change in males
were low, and that it seems difficult to conclude that one or the other sex is
preferentially impacted, suggesting that your results are not so convincing. This is
not so motivating for the reader then to follow this subsequent discussion 

- Line 627-628: these observations could have indeed important consequences with
respect to drug treatment, but again, how serious do we have to take this, and why
would only S. haematobium males in heterospecific pairing have this response, and
not S. bovis males?

- Line 632: avenues

- Line 634: but these report unidirectional introgression of a few bovis genes into
the haematobium genome, so a dominance of haematobium genomic DNA, how can you
reconcile or link this with your results? Wouldn’t you expect more differences
between the different crosses?

It has been proven in co-infection experiments that crosses are not always
reciprocal, i.e. one cross performing better than the reverse cross. This could also
be expected in these crosses as in Senegal many hybrids that were found appeared to
have arisen of a female S. bovis x male S. haematobium pairing (although
backcrossing in nature obscures these patterns, and in other areas, like Niger, the
reverse crosses are more frequently found). I do not see any reference to this or to
the topic of unidirectional hybridisation and unidirectional introgression, which I
think is missing.

- line 648: I don’t completely agree, because the different ‘compartments’ (strange
term) do not prevent them from meeting each other in the liver, where mating takes
place, only after that stage the couple moves through the hepatic portal vein to the
egg-laying site. So there is plenty of opportunity in the liver, irrespective of the
difference in tropism

- line 653: how can these parasites be co-occurring (I would use the word sympatric)
of their hosts are not? Please adapt

- line 657: rephrase ‘parental species individuals’

- line 655-659: what do the authors want to say here exactly? Does ‘such hybrids’
refer to those hybrids resulting from an ancient hybridisation event? And do only
‘those hybrids’ present heterosis, in contrast to ‘other, more recent hybrids’?
Please rephrase. Also, how do your results explain the fact that previous studies
show that mainly ‘ancient hybrids’ are found in nature (Platt et al, 2019), while
your experiments show that hybridisation is so ‘easy’? In line 668 you write that
your result ‘echoes recent evidence of introgression’. I am not sure what you mean
with this. Do you mean ‘evidence of recent introgression’ or ‘recent evidence of
(ancient) introgression’? I would say the opposite, the high prevalence of hybrids
is an echo or a reflection of the weak pre-zygotic barrier that you observed. But
still, I don’t see how your results can be matched to the outcome of Platt et all,
suggestion ancient hybridisation.

- Line 669: a higher

- Line 670: if the hybrids always have higher fitness, why do you still see ‘pure S.
haematobium’ and ‘pure’ S. bovis in places where they overlap? Wouldn’t the hybrids
take over? 

- Line 672: ‘new issues in the disease control’ and ‘alteration in the efficacy’
sound rather vague as a closing sentence, please be more specific.

Reviewer #3: Summary

The goal of this work appeared to be to test for evidence of species-specific mate
choice between S. haemotobium and S. bovis, and also to see whether there are
differentially expressed genes in adults engaged in homo vs. hetero-specific
pairings. The found minimal evidence for mate choice and few strongly DE genes. In
the discussion they speculate on possible roles of the few DE genes, but make no
strong conclusions about any of them. 

Comments

The authors did a mate choice experiment using cercariae of known sex in hamsters.
They found no strong evidence that species-specific mate choice occurs by males or
females of either species. The statistical analysis of the mate choice experiments
seemed appropriate to me. 

 If they have the data, it would be interesting if the authors could comment on the
physical location of the hetero vs homo-specific pairings within the hamster
(urogenital vs. mesenteric). I wondered whether different behavior might be observed
if the definitive host was a larger mammal such as a bovine or human. Perhaps the
authors would care to speculate or at least mention the possibility.

The authors also looked for evidence of differential gene expression in individuals
of each sex and species when engaged in hetero vs. homo-specific pairings. They
observed only a few dozen DE genes in females of either species. Oddly, they found
zero DE genes in male S. bovis, but over 1000 in male S. haemotobium (although I
wonder if figure 2a suggests the difference in S. haemotobium might be driven by one
individual). 

The almost complete lack of even minimally DE genes in S. bovis males shown in figure
3d is puzzling. I have never seen a volcano plot like this one. I would have
expected more by chance alone with only n = 3 per treatment. Analysis of gene
expression data is not my expertise, so I defer to other reviewers to comment on
this. Or perhaps the authors could head off puzzled readers by explaining why this
pattern obtains.

Minor comments:

The manuscript could use some editing to fix various small grammatical errors and
instances of odd English usage. Perhaps asking a native English speaker to read it
through once for them would be helpful. 

It would help Figure 3 if the authors would label, on the figure, which combination
of sex and species is represented by each panel. Going back and forth between the
legend and the figure is tedious for the reader. 

Figure 4. Is there some measure of statistical significance associated with the
difference between blue and red bars that could be indicated on the figure?

Supplementary table S1 would be helped by species identifications.

PLOS authors have the option to publish the peer review history of their article
(what does this mean?). If published, this will
include your full peer review and any attached files.

If you choose “no”, your identity will remain anonymous but your review may still be
made public.

**Do you want your identity to be public for this peer review?** For
information about this choice, including consent withdrawal, please see our
Privacy Policy.

Reviewer #1: No

Reviewer #2: No

Reviewer #3: No
---

## [Decision Letter · Decision Letter 1]

2 Feb 2021

Dear Mrs Mathieu-Bégné,

Thank you very much for submitting your manuscript "No pre-zygotic isolation
mechanisms between Schistosoma haematobium and Schistosoma bovis parasites: from
mating interactions to differential gene expression" for consideration at PLOS
Neglected Tropical Diseases. As with all papers reviewed by the journal, your
manuscript was reviewed by members of the editorial board and by several independent
reviewers. The reviewers appreciated the attention to an important topic and we all
agree that the manuscript is now greatly improved. Based on the reviews, we are
likely to accept this manuscript for publication, providing that you modify the
manuscript according to the review recommendations. I look forward to seeing an
updated paper resubmitted very soon.

Sincerely,

Poppy H L Lamberton

Deputy Editor

Poppy Lamberton

Deputy Editor

The manuscript is now greatly improved and I look forward to seeing these minor
changes suggested by two of the reviewers amended and an updated paper resubmitted
very soon.

Reviewer's Responses to Questions

**Key Review Criteria Required for Acceptance?**

**Methods**

-Are the objectives of the study clearly articulated with a clear testable hypothesis
stated?

-Is the study design appropriate to address the stated objectives?

-Is the population clearly described and appropriate for the hypothesis being
tested?

-Is the sample size sufficient to ensure adequate power to address the hypothesis
being tested?

-Were correct statistical analysis used to support conclusions?

-Are there concerns about ethical or regulatory requirements being met?

Reviewer #1: The methods are now clear and easy to follow

Reviewer #2: yes

Reviewer #3: Acceptable

**Results**

-Does the analysis presented match the analysis plan?

-Are the results clearly and completely presented?

-Are the figures (Tables, Images) of sufficient quality for clarity?

Reviewer #1: The results are clearly presented.

Reviewer #2: yes

Reviewer #3: Acceptable

**Conclusions**

-Are the conclusions supported by the data presented?

-Are the limitations of analysis clearly described?

-Do the authors discuss how these data can be helpful to advance our understanding of
the topic under study?

-Is public health relevance addressed?

Reviewer #1: The discussion supported the data presented and is informative. The
limitations are clearly presented and there relevance to public health is
addressed.

Reviewer #2: yes

Reviewer #3: Acceptable

**Editorial and Data Presentation Modifications?**

Reviewer #1: See general comments

Reviewer #2: The authors did a good job in addressing all the comments of the
referees, they adapted the text and figures/tables where needed or suggested, and
this resulted in a much-improved version of the manuscript. I think this version can
be accepted for publication.

A few minor comments:

Line 43-44: before reproduction is also ‘during the life cycle’, so why not simply
saying ‘These preventive barriers can act before reproduction, “pre-zygotic
barriers”, or after reproduction/fertilisation, “post-zygotic barriers”?

Line 561; draw

Line 569: male transcriptomes

Line 640; to overcome their divergence only I would say, not to overcome their
relatedness

Line 654: a lower sensitivity to PZQ of the hybrid being at the origin of the spread
was not at all proposed by Huyse et al., 2009 so please adapt. 

Line 708: S. mansoni [28]).

Line 711; I think the biased patterns in the field might indeed by due to
post-zygotic isolation but not the rare encounter of first generation hybrids (or
not only because of that). The latter is probably (also) due geographical variation
in the level of sympatry between parental species, no?

Reviewer #3: They adequately addressed the issues of language usage.

**Summary and General Comments**

Reviewer #1: The paper is much improved, and the authors have taken the time and
effort to amend the manuscript according to the reviewer’s comments. This is a
highly informative paper that adds to our understand on the interactions between S.
haematobium and S. bovis. It will also open several research avenues for future
work. 

I have the following minor comments. 

Gramma punctuation is needed in several places so a good go over will help

Some further modifications of the English is needed. Areas that I managed to pick up
are highlighted. 

Line 120 – there are 23 species not 25 

Line 121 – to be precise 20 infect animals if you include S. mansoni that is found in
rodents and monkeys. 

Line 188-189 – make it clear that the heterosis has been observed in experimental
infections not in the natural setting. 

Line 220-222- the animal license information should be checked with the editors
regarding how it should be written. 

Line 229 – what do you mean by sympatric B. truncatus – do you mean local natural
hosts? 

Line 250 – add “below” to the end of the sentence. 

Figure 2 – S. h = S. bovis needs correcting. Also, the symbols are both in the figure
and legend. Only one is needed. 

Table 1: in Exp. 2 and 4 you say that this is a female choice experiment. Is this not
more competition of the two male species?

Line 290 – make it clear that this section is moving onto the molecular work and not
the analysis of pairings. 

Line 296 – make it clear how the worms were separated. It is important here that
there is not contamination between males and females, so the detail is needed. 

Line 327 – “effect” not needed 

Line 328 – reference the Galaxy instance. E.g., is it a software or a machine? 

Table 2 column 6 and 7 remove the words or change homo specific or hetero specific as
they are not paired. Thy can just be called unpaired worms. It would also be better
to have the exp. row above the data rather and below. 

For figure 4 is it important to distinguish between hetero and homo paired worms ? 

Line 481 – show what GO means, 

Line 553 – is there any evidence for female selecting their partners. They are very
underdeveloped until they are paired? 

Line 555 – this S. intercalatum strain is actually now named S. guineensis – maybe
add a note on this and also where it is referenced in the intro. 

Line 565 – reference the study Webster et al., 2012 that also suggested that there
was no mate choice 

Line 599 - suggest you add here that the majority of the hybrids found in the field
appear to be a result of a cross between S. h male and S. b female based on cox1 and
ITS sequencing. It is better to say this than state that they are as more work is
needed to clarify that. 

Line 655 – I think it is important to point out that any changes in PZQ response by
hybrids is very theoretical and there is not current evidence that there is any
difference in drug response in natural infections. 

Line 707- check if the S. intercalatum should be called S. guineensis. S.
intercalatum is the Zaire/DRC strain. Cameroon and other areas is S. guineensis.

Reviewer #2: (No Response)

Reviewer #3: I believe the authors have adequately addressed my comments on their
original submission. The main points of the paper, that there is random mating and
few differentially expressed genes in interspecies pairings, are sufficiently
supported and worth putting into the literature

PLOS authors have the option to publish the peer review history of their article
(what does this mean?). If published, this will
include your full peer review and any attached files.

If you choose “no”, your identity will remain anonymous but your review may still be
made public.

**Do you want your identity to be public for this peer review?** For
information about this choice, including consent withdrawal, please see our
Privacy Policy.

Reviewer #1: Yes: Dr Bonnie Webster

Reviewer #2: No

Reviewer #3: No
---

## [Editor Report · Decision Letter 2]

6 Apr 2021

Dear Mrs Mathieu-Bégné,

We are pleased to inform you that your manuscript 'No pre-zygotic isolation
mechanisms between Schistosoma haematobium and Schistosoma bovis parasites: from
mating interactions to differential gene expression' has been provisionally accepted
for publication in PLOS Neglected Tropical Diseases.

Best regards,

Poppy H L Lamberton

Deputy Editor

Poppy Lamberton

Deputy Editor

There is some minor type editing required, such as:

Line 160: remove full stop and close bracket.

Line 231: remove space at start

Line 233: I think you can revert to sympatric here, I think it is clear, but maybe if
clarification is needed write 'and sympatric B. truncatus molluscs, bred from snails
collected from the same location as the parasites, were individually .....' and
please calrify here also, or if these were collected straight from the field and
exposed, then remove the 'bred from'

Line 376: Please add 'worms of the limiting sex (i.e., choosing partners, such as
female choice or male competition) and please include this in the table legend as
well to explain it when it is first used.

Several references need the Latin names in italics and the titles having the full
capitalisation removed

---

## [Editor Report · Acceptance letter]

29 Apr 2021

Dear Mrs Mathieu-Bégné,

We are delighted to inform you that your manuscript, "No pre-zygotic isolation
mechanisms between Schistosoma haematobium and Schistosoma bovis parasites: from
mating interactions to differential gene expression," has been formally accepted for
publication in PLOS Neglected Tropical Diseases.

Best regards,

Shaden Kamhawi

co-Editor-in-Chief

Paul Brindley

co-Editor-in-Chief
